# Automated Defect Analysis of Additively Fabricated Metallic Parts Using Deep Convolutional Neural Networks

**Saber Nemati [1],\*, Hamed Ghadimi [1], Xin Li [2], Leslie G. Butler [3], Hao Wen [1] and Shengmin Guo [1]**

1    Department of Mechanical and Industrial Engineering, Louisiana State University,
     Baton Rouge, LA 70803, USA
2    Department of Visualization, Texas A&M University, College Station, TX 77843, USA
3    Department of Chemistry, Louisiana State University, Baton Rouge, LA 70803, USA
\*    Correspondence: mnemat2@lsu.edu

**Abstract:** Laser powder bed fusion (LPBF)-based additive manufacturing (AM) has the flexibility in fabricating parts with complex geometries. However, using non-optimized processing parameters or using certain feedstock powders, internal defects (pores, cracks, etc.) may occur inside the parts. Having a thorough and statistical understanding of these defects can help researchers find the correlations between processing parameters/feedstock materials and possible internal defects. To establish a tool that can automatically detect defects in AM parts, in this research, X-ray CT images of Inconel 939 samples fabricated by LPBF are analyzed using U-Net architecture with different sets of hyperparameters. The hyperparameters of the network are tuned in such a way that yields maximum segmentation accuracy with reasonable computational cost. The trained network is able to segment the unbalanced classes of pores and cracks with a mean intersection over union (mIoU) value of 82% on the test set, and has reduced the characterization time from a few weeks to less than a day compared to conventional manual methods. It is shown that the major bottleneck in improving the accuracy is uncertainty in labeled data and the necessity for adopting a semi-supervised approach, which needs to be addressed first in future research.

**Keywords:** laser powder bed fusion; X-ray computed tomography; image segmentation; machine learning; deep learning

## 1. Introduction

In the Industry 4.0 era and with the emergence of additive manufacturing (AM) and digital twins, data-driven material development and manufacturing process optimization has become possible due to the abundance of data. The workflow of a typical metal AM optimization process includes phase diagram calculations (CALPHAD), feedstock production, part fabrication, non-destructive evaluation, and microstructure and mechanical testing. Each of these stages has sets of parameters that are inter-correlated and determine the final properties of the fabricated part. For example, in laser powder bed fusion (LPBF)-based AM, the combination of alloy compositions and AM processing parameters (laser power, laser spot diameter, scan speed, layer thickness, and hatching pitch) will impact the fatigue life [1] and other mechanical properties.

Finding the correlation between the parameters throughout the process can help researchers simulate and adjust them in a controlled way to obtain superior material properties. However, one of the biggest challenges is handling the "huge amount of data". Nowadays, the most feasible solution to this problem is AI-assisted data handling and data processing. This can be applied in all stages of the data-driven model. The focus of this paper is on how to handle non-destructive evaluation (NDE)-generated data, and in particular, segmentation and characterization of tomography data for automated defect detection for metallic parts fabricated by LPBF.

Image segmentation is introduced as partitioning an image into its regions based on some criteria where the regions are meaningful and disjoint [2]. It is traditionally performed by an expert (the so-called manual segmentation), which is a cumbersome task due to the large amount of data. As a result of rapid developments in machine learning (ML), many model-based algorithms have been introduced for automatic image segmentation. In this approach, a trainable architecture (i.e., AI agent) learns how to perform a certain task that is normally handled by a human expert.

A key factor in interpretation of the experimental X-ray imaging data is segmentation into images with information content close to ground truth. Different types of internal defects with various morphologies may be distributed throughout the part. Therefore, in order to increase the possibility of capturing all different types of defects, a limited number of images from different sections of the sample that are far away from each other should be chosen and manually segmented by an expert and approved by other experts. Due to the difficulty of manual image segmentations, any practice that minimizes the use of manually segmented images for training and populates the training dataset using data augmentation methods is extremely valuable. This approach is also known as *few-shot learning* in the context of computer vision.

A tomography dataset is often four-dimensional, with three-dimensional space plus a time dimension or environmental variable. Herein, we have extracted a handful of representative 2D slices for training the automated image segmentation workflow, a workflow that is then applied to a 3D volume.

The reconstructed features of the part can be used for predicting and analyzing the mechanical behavior of the manufactured part under different loading conditions (e.g., static, cyclic, etc.). Specifically, the fatigue life of AM parts strongly depends on the morphology of the cracks and microcracks on the surface and inside the parts. An accurate understanding of those features can greatly contribute to accurately predicting fatigue life. Therefore, the accurate identification of internal features is the most important objective of this study.

The automated segmentation procedure can be improved in different stages, including but not limited to the following:

- Selective manual segmentation: Acquiring manually segmented images are costly. Therefore, the images that are selected to be manually segmented by the domain experts should contain information that is likely to be found in the entire dataset. For example, if the scanned sample contains pores and cracks, the selected training slices should have different types of pores and cracks and their combination to use human expertise as much as possible [3].
- Data augmentation: Methods such as cropping, rotating, and adding noise that enable populating the training dataset with a limited amount of distinct labeled data [4,5]. This technique also improves the network performance in terms of robustness against noises and lowers the chance of overfitting.
- Network architecture: Attempts to design more efficient and more intelligent network architectures to get close to—or even outperform—the human brain are categorized under this stage. Most of the recent papers have focused on this area during the last few years [6,7].
- Evaluation measures: The criteria that determine how far (close) the network output is from the ground truth. The optimizer tries to minimize (maximize) these criteria during the training phase [8].

Although not all the attempts in improving automated image segmentation and defect analysis fall under these categories, this framework allows one to focus on one aspect of the workflow and study the effect of one module while keeping other modules unchanged.

In the context of defect analysis in material science and manufacturing processes, the challenge of "unbalanced class", in which the number of pixels corresponding to defects is considerably lower than that of background, normally arises. These small classes, despite their small share in the entire image, are highly influential on the mechanical properties, in

particular fatigue strength. In other words, cracks may be present in 1% of the image pixels in typical material science datasets, but they drastically dictate the mechanical performance of the specimen, and one cannot afford to leave them unidentified. Therefore, the correct and reliable quantification of those small regions can play a major role in predicting material strength and finding their correlation with processing parameter reliably.

Several categories of metals and alloys have been widely investigated with the LPBF process for high printing quality and a very low number of defects, such as Ti-6Al-4V [9], Inconel 718 [10], Inconel 625 [11], stainless steel 316L [12], and CuCrZr alloy [13]. As the focus of this paper is on automated defect analysis of AM parts using deep convolutional neural networks, samples with a measurable number of defects should be prepared. In this regard, Inconel 939 alloy was used in this paper as the feedstock material for the LPBF process. Inconel 939 alloy has many outstanding properties, i.e., high creep resistance, excellent corrosion/oxidation resistance, and high-temperature microstructure stability. The primary cracking mechanism for Inconel 939 alloy is solidification cracking, which usually occurs close to the end of the solidification stage as the liquid feed in the inter-dendritic region is constrained [14]. Therefore, Inconel 939 is generally regarded as a non-weldable alloy and it is expected to have a large number of defects in the LPBF parts.

Many researchers have attempted to address the automated segmentation problem in the context of material science and manufacturing processes [5–7,15–23], but most of them were focused on designing novel network architectures. Few statistical investigations and sensitivity analyses have been conducted to reveal the utmost capability of simpler architectures for a typical X-ray CT scan of an LPBF part with unbalanced classes of pores and cracks. Consequently, one must objectively justify the need for using different configurations of simple architectures for a given task, before moving forward with more complicated architectures.

U-Net is a widely used baseline pipeline for image segmentation; it has a simple architecture that nicely balances accuracy, robustness, and efficiency. Therefore, the U-Net architecture is chosen for detailed analysis herein. U-Net is the basis for many new architectures for image segmentation [24–32]. The goal of this paper is to comprehensively identify any unique situations that either U-Net cannot handle or that need to be addressed in a different way other than using more complicated architectures. We examine different settings of U-Net architecture to gain a deep understanding of their capability in capturing small features, in particular crack tips, microcracks, and micropores. Moreover, this paper tries to identify the bottlenecks throughout the process and prioritize them to be addressed later in future research. This approach helps researchers to focus on improving the most influential stages of the process, instead of adopting a trial and error strategy for maximizing the segmentation accuracy. The bottlenecks that are investigated in this paper are:

- Effect of network depth
- Random weight initialization
- Accuracy of the manually labeled data compared to network prediction

As we develop an understanding of the shortcomings of the current automated defect segmentation process with U-Net, we are in a better position to pinpoint the critical stages and assess the strengths and weaknesses of the new image segmentation architectures, if using a more complicated one is ever needed.

## 2. Materials and Methods

### 2.1. Material and Fabrication

Spherical Inconel 939 alloy powders were supplied by LPW Technology, Inc, which were gas-atomized with the particle size of 15–45 microns; the chemical composition is listed in Table 1. The cylindrical-shaped LPBF samples were prepared using a Concept-Laser Mlab cusing-R system (Lichtenfels, Germany) in argon atmosphere with the residual oxygen level under 0.2%. The processing parameters are as follows: laser power 95 W (continuous), scanning speed 100 mm/s, hatch spacing 50 μm, layer thickness 25 μm, and scan strategy "with islands" with island size 5 mm × 5 mm. The islands have a

shift of 0.2 mm and the scan direction has a rotation angle of 90 degrees between adjacent layers. This scan strategy is adopted to reduce the residual thermal stress inside the sample during fabrication process. Figure 1a shows a schematic diagram of the scan strategy. The height and diameter of the final printout cylindrical-shaped LPBF samples were 30 mm and 14 mm, respectively, as shown in Figure 1b. The cylinders were cut off from the build platform using a wire electrical discharge machine (EDM).

**Table 1.** Elemental composition of Inconel 939 powders.

| Elements | Al | B | C | Co | Cr | Fe | Mg | N | Nb | Ni | Si | Ta | Ti | W | Zr |
|---|---|---|---|---|---|---|---|---|---|---|---|---|---|---|---|
| Contents (wt.%) | 1.9 | 0.01 | 0.15 | 19.2 | 22.3 | 0.1 | <0.1 | 0.01 | 1.0 | Bal. | 0.1 | 1.5 | 3.6 | 2.0 | 0.11 |

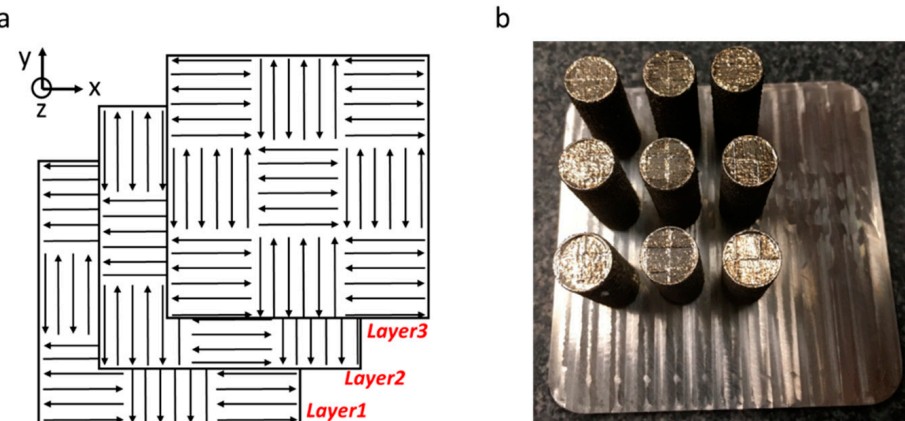

**Figure 1.** (**a**) Schematic of island laser scan strategy; (**b**) image of the LPBF Inconel 939 samples.

### 2.2. Phase Constituents and Microstructures

A 3 mm thick disk was cut off from the LPBF samples for material characterization. The surface of the disk sample was mechanically ground using SiC papers of different grit sizes (400, 600, 800, 1000, and 1200 grits, successively), then polished with the MetaDiTM Supereme polycrystalline diamond suspension (6 μm, 3 μm, 1 μm, in sequence). After polishing, the sample surfaces were etched using etching solution, which was nitric acid and hydrochloric acid mixture with a volume ratio of 1:3. Finally, the microstructures of sample surface were characterized with a Quanta™ 3D DualBeam™ FEG FIB-SEM scanning electron microscope (SEM).

Figure 2 shows the SEM images of the LPBF Inconel 939 with different magnifications. All the images were taken perpendicularly to the building direction. For as-fabricated Inconel 939 samples, equiaxed cellular structure was observed with a size of around 1 μm, as shown in Figure 2a,b. Similar structures were also found by other researchers for 3D-printed nickel-based alloys. In many cases, such microstructure leads to considerably higher strength due to the Hall–Petch effect. White particles are also observed along the cellular structure boundaries, and it was claimed that these partials are carbide phases [33,34]. In the as-fabricated Inconel 939 samples, defects such as pores and cracks ranging from a few microns to hundreds of microns are obvious (Figure 2b,c), which indicates the poor quality of the LPBF samples. It is worth noting that besides the cracks and sharp-corner-shaped defects, spherical pores with the size of several tens of microns (indicated by white arrows in Figure 2c) are also visible in the sample, which are most likely the key-hole voids caused by excess laser energy input. Similar phenomenon was also reported elsewhere [35]. Archimedes principle was applied to measure the densities of the samples. Compared with casted Inconel 939 (8.17 g/cm$^3$), the porosity of the LPBF sample is estimated to be 3.72%.

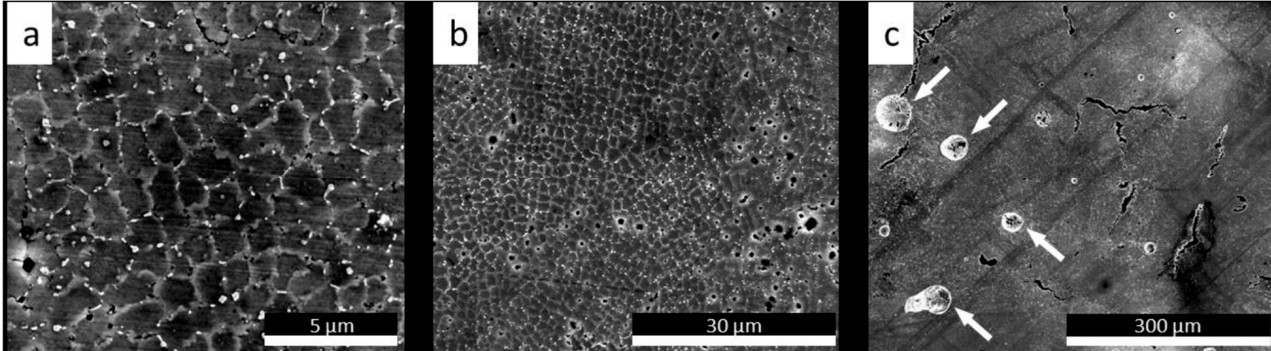

**Figure 2.** SEM images showing the microstructures of the LPBF Inconel 939 with different magnifications. (**a**) grain structure (**b**) the distribution of defects within the grain structure (**c**) morphology of the cracks and the pores.

To understand the phase compositions of Inconel 939, CALculation of PHAse Diagrams (CALPHAD) calculations were performed using Thermo-Calc 2022b software package with TCNI8 database (Ni-Alloys v8.2). Figure 3 shows the CALPHAD calculation results, which describe the relation between temperatures and the amount of all phases at equilibrium states. It can be found that at 800 °C, the main phases are γ (FCC_L12), γ′ (FCC_L12#2), σ (SIGMA), η (NI3TI_D024), and carbide phases (M23C6); at 1200 °C, the main phases are γ (FCC_L12), γ′ (FCC_L12#2), carbide phases (FCC_L12#3).

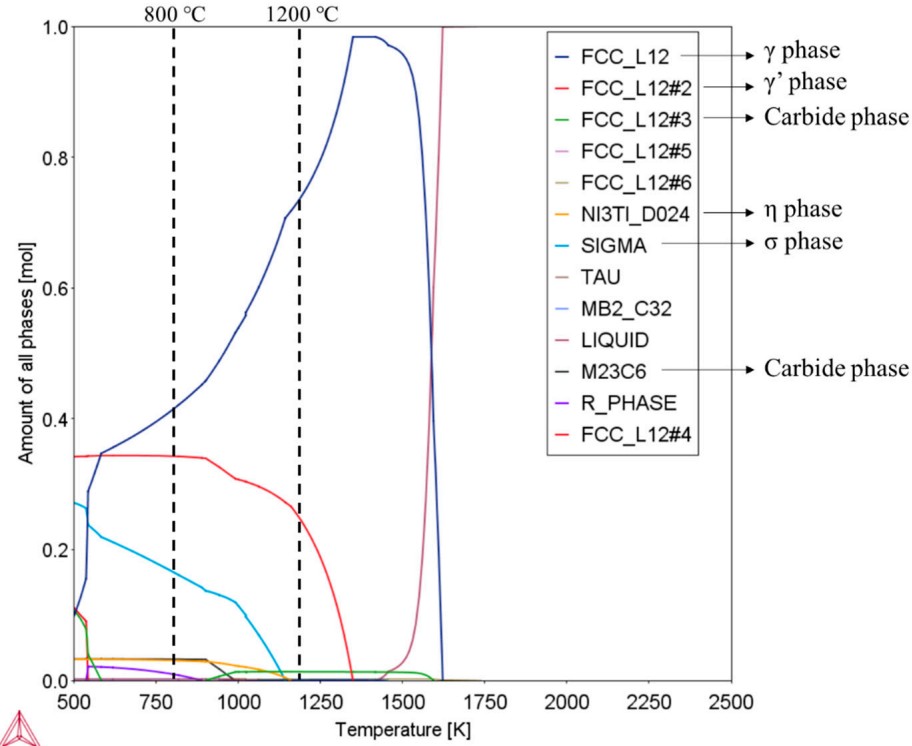

**Figure 3.** The number of different phases at different temperatures for Inconel 939 alloy obtained from CALPHAD calculation.

## 3. CNN-Based Architecture

In recent years, artificial intelligence (AI) and machine learning (ML) have shown promising performance in data analysis. In particular, convolutional neural networks (CNNs) have been very effective in computer vision tasks. The early documented evidence of using convolutional neural networks for image classification includes papers published

by Fukushima [36] and LeCun et al. [37] back in the 1980s. Since then, CNN-based architectures have been the de facto method for image segmentation and recognition tasks due to their ability in object localization [38–45]. After the introduction of GPU training by Krizhevsky et al. [46], there has been a giant leap both in deep network architecture training and the GPU production industry.

In the context of dense prediction and semantic segmentation, a specially designed architecture was branched out of fully convolutional networks (FCNs) [42] for medical image applications: U-Net, introduced in 2015 by Ronneberger et al. [47], has shown great performance in biological image segmentation by aggregating the low-level and-high level features using an encoder–decoder scheme along with skip connections. Many other architectures, including 3D U-Net [24], V-Net [25], Unet$^e$, Unet+, Unet++ [26], and H-DenseUNet [27], have inherited from the idea behind U-Net and further improved the test set performance, mostly in medical image segmentation tasks.

U-Net architecture [47] has a contracting path as the encoder and a symmetric expanding path as the decoder. The encoder extracts the contextual information at different resolutions and the decoder localizes the features based on the information that the encoder has abstracted. A segmentation head—which is typically a fully connected layer—is included at the end to map the decoder output to the segmentation map. The encoder layers can be either as simple as plain convolutional layers with activation or deep ViT-based [28,29,48] architectures. In this context, the encoder is also known as the *backbone*. A five-level deep U-Net architecture with a ResNet18 encoder (backbone) is shown in Figure 4.

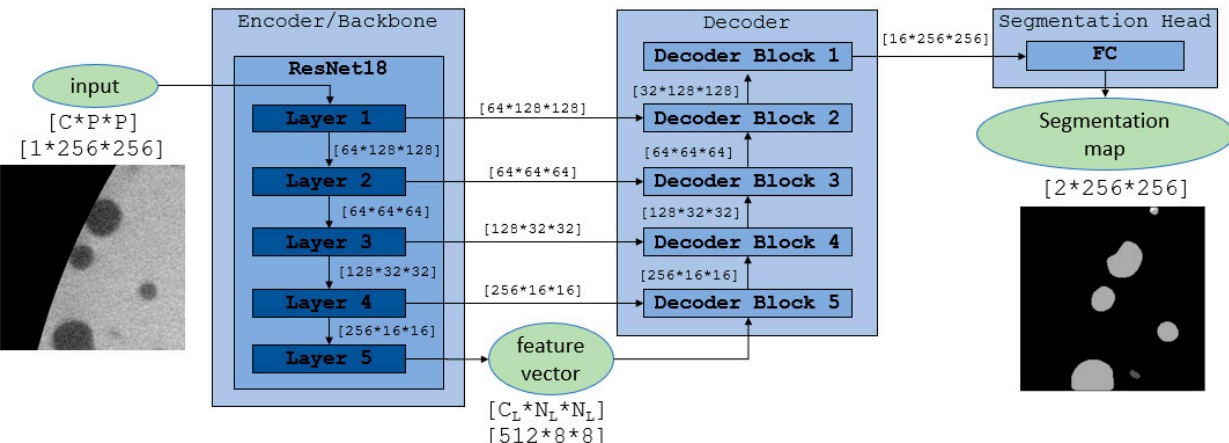

**Figure 4.** U-Net architecture with ResNet18 backbone and standard U-Net decoder. The encoder follows the exact same architecture of ResNet18 available in [49]. The dimensions at each resolution are calculated based on a grayscale 256 × 256 patch of the original image.

While simple and shallow backbones with a fewer number of trainable parameters can significantly reduce the computational cost in the training phase, they may not be able to capture all the contextual information of the input and reduce the overall performance. On the other hand, heavier and overly complicated backbones will increase the computational cost with little noticeable improvement. Moreover, using more complicated networks with more trainable parameters always makes the model exposed to *overfitting*. Therefore, a trade-off should be made for maximum efficiency.

One of the ultimate quests is to find a correlation between the physics behind the dataset derived from the actual sample and the network settings in a way that can facilitate architecture tuning. For example, the average size and the morphology of the defects inside the specimen should be a good clue in choosing the optimum patch size during the data augmentation process. On the other way around, an optimum set of network parameters may help in understanding the physics behind the fabrication process.

U-Net downscales the resolution of the original image using 2D convolutional filters to a certain depth/level (5 in case of Figure 4). A pure downscaling with identity filters is illustrated in Figure 5. This downscaling is beneficial in focusing on detecting global features (e.g., background vs. base metal), because almost all the local features are filtered out at the lowest resolution and only global ones are identifiable.

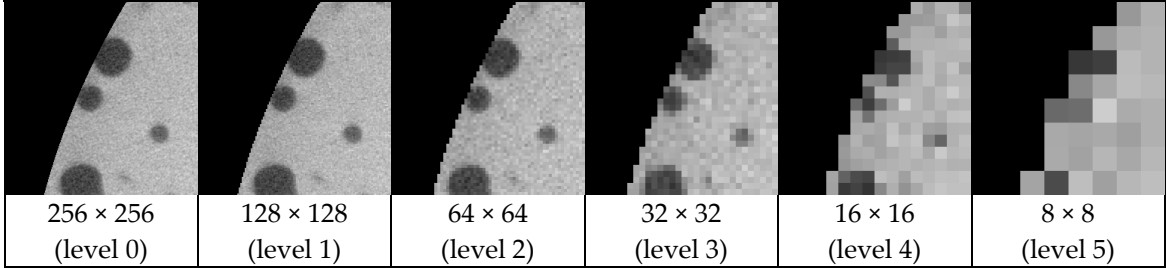

| 256 × 256 (level 0) | 128 × 128 (level 1) | 64 × 64 (level 2) | 32 × 32 (level 3) | 16 × 16 (level 4) | 8 × 8 (level 5) |

**Figure 5.** Pure downscaling of a patch. This figure illustrates the resolution at each level of U-Net.

## 4. Automated Defect Analysis

Following the supervised learning approach, two images—which include all the features and are shown to be sufficient for training in the results section—are manually segmented by an expert. Namely, each pixel of the images is assigned to a class label corresponding to a specific feature (i.e., background, pore, and crack) based on the expert's point of view. Next, the data are converted into adequate multidimensional arrays (tensors) that the classifier can understand. These two phases are addressed separately.

### 4.1. Manual Image Segmentation

Three-dimensional X-ray computed tomography (X-ray CT) data on the LPBF Inconel 939 sample were collected by using the ZEISS Xradia 620 Versa instrument. The scanning volume is a cylinder 1.3 mm in height and 1.4 mm in diameter contained within the cylindrical-shaped sample reported under Section 2. The scanning resolution is 0.7 microns per cubic voxel. Each tomographic slice is a 16-bit grayscale image of size 2005 × 2043 pixel, and 1879 slices along the height of the cylindrical volume were extracted. The slices were later converted into 8-bit images for memory management, as little difference between the 8- and 16-bit depths was observed in the preliminary results. The sample includes two different categories of defects as judged by the domain expert: pores and cracks. Pores are spherical-shaped voids (circular shapes in 2D slices) inside the sample, possibly created by excessive laser power, also known as keyhole voids [35]. Cracks are irregularly shaped planar features (jagged lines in 2D slices) that are likely to be formed due to thermal stresses during laser powder bed fabrication. An overview of the scanned volume is shown in Figure 6.

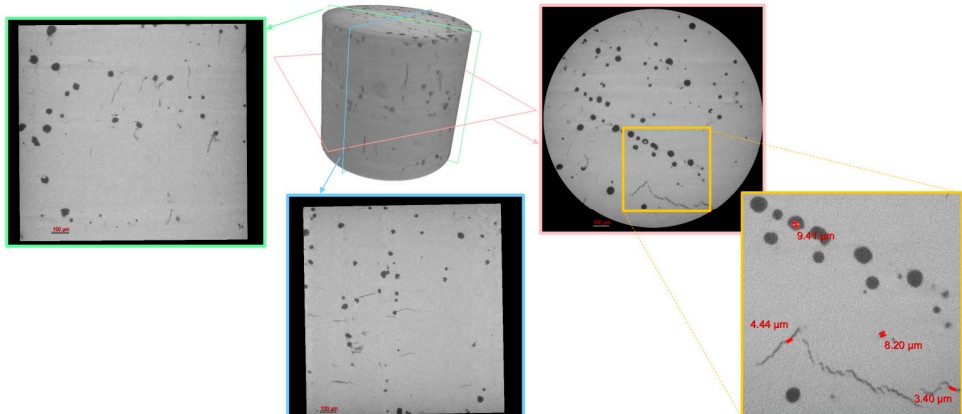

**Figure 6.** Overview of the scanned sample using ZEISS Xradia 620 Versa.



We manually segment the original 2D image by simply brushing the pixels that are identified as pores and cracks with a specific color—in this case, Magenta (RGB: 255, 0, 255). The original image and the manually segmented images are shown in Figure 7.

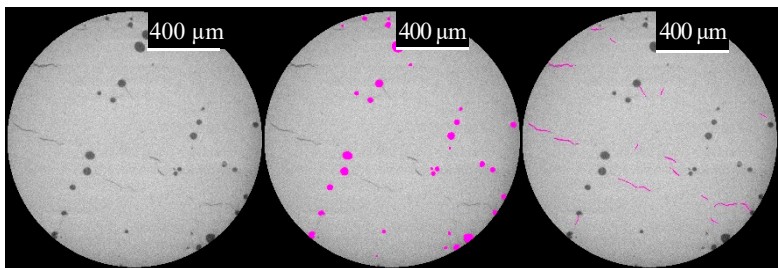

**Figure 7.** ZEISS sample segmented image. **Left**: original image, **Center**: pores, **Right**: cracks.

### 4.2. Data Preparation

From a total of 1879 2D slices, two grayscale 8-bit 2005 × 2043 2D images are considered for populating the training and validation sets, and another image from a different section of the same sample is chosen for populating the test set and evaluating the performance of each architecture. The selected images contain all the color contrast and morphological features that need to be learned by the network. For populating the datasets, a simple data augmentation method is used: each image is divided into 64 patches, each with a size of 256 × 256 pixels. Preliminary test runs show that this patch size can encompass the morphological characteristics of both crack and pore features. A larger patch size will not produce enough training data, while very small ones will not be able to represent the features properly. For splitting the original image into non-overlapping equisized patches, the black background is extended in such a way that the dimensions are dividable by the patch size without losing information. A sample image with selected patches is illustrated in Figure 8. The same procedure is repeated for the segmentation map, and each patch will have a corresponding manually segmented binary map, which is called a *label* (Figure 9). With the proposed data preparation scheme, input and output dimensions are [*B*, *C*, *P*, *P*] and [*B*, *1*, *P*, *P*], where B, P, and C are *batch size*, *patch size*, and *number of channels*, respectively.

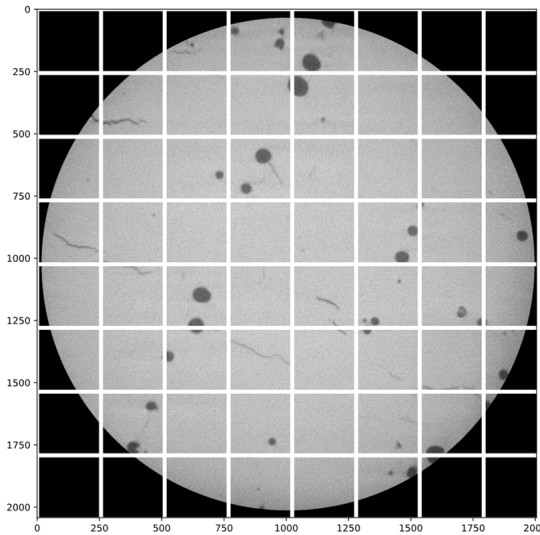

**Figure 8.** The 256 × 256 patches used as training set (number of patches: 64).

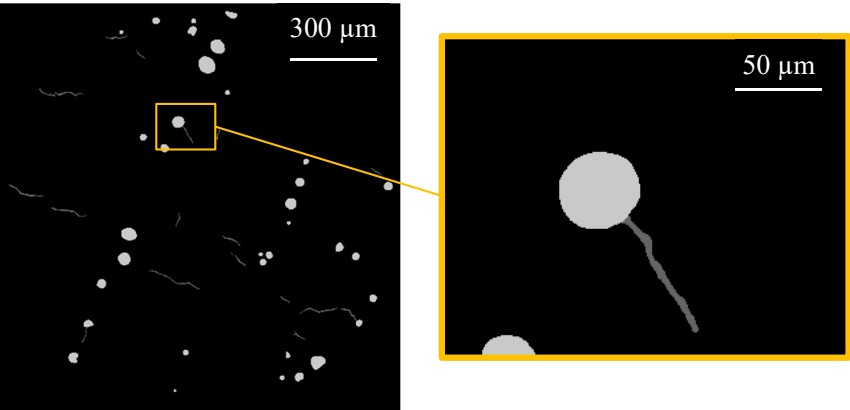

**Figure 9.** A training segmentation map or *label*.

### 4.3. Performance Metric

To put more emphasis on the correct classification of defects rather than the background, the Dice coefficient is used as the loss function and the intersection over union criteria (also known as the Jaccard index) is used for evaluating training performance [8].

$$Dice = \frac{2TP}{2TP + FP + FN} \tag{1}$$

$$IoU = \frac{TP}{TP + FP + FN} \tag{2}$$

where *TP* is the number of true-positive pixels, *FP* is the number of false-positive pixels, and FN is the number of false-negative pixels. Accordingly, the *IoU* of the ground-truth map is 1, and any deviation from it will move it closer to zero. The mean value of *IoU* is also considered a good representation of the overall performance of the network in all categories.

$$mIoU = \frac{IoU_{class\ 1} + \ldots + IoU_{class\ (n)}}{n} \tag{3}$$

The Dice coefficient (F1 score) is another form of *IoU* score with the same range and behavior.

### 4.4. Training

There are many settings that affect the overall performance of the network, including:

- Data augmentation settings: patch size, normalization;
- Network settings: depth, backbone, layer structure, decoder structure, segmentation head, normalization, regularization;
- Training settings: learning rate, size of training/validation/test set, optimizer, loss function, weight initialization.

Random weight initialization is an effective method in minimizing the possibility of getting stuck into a local minimum. Adding this randomness to the model will lead to a different set of trained weights for each run. Therefore, each model is trained 10 times, and the distribution of the calculated mIoU on the test set is reported.

Preliminary results showed that the ResNet18 backbone yields better mIoU compared to more complicated backbones (such as ResNet34 and ResNet50). Therefore, to establish conditions for a fair comparison, "ResNet18" pre-trained on the ImageNet dataset is chosen as the encoder (also known as backbone). No augmentation method other than regular non-overlapped patching is used for populating the training/validation/test sets. Two manually labeled images are used for training and validation and one manually labeled image is used for testing, each of which is taken from a different section of the specimen. The effect of other network settings is left for future research. The Segmentation

Models Pytorch [50] package is used for implementing the architectures and the developed repository can be accessed using the following GitHub link, accessed on 21 October 2022: https://github.com/LSU-LAMDA/crack-pore-detection.

## 5. Results and Discussion

The overall performance of each setting is measured by the average intersection of union (mIoU) of the resulting segmentation map on the test set. The time that it takes to train the network with a single NVIDIA GeForce GTX 1650 GPU is also reported as a measure of computational cost. ResNet18 has exactly 11,176,512 trainable parameters, and since it is the backbone for all architectures, the total number of trainable parameters will only differ on the decoder side and segmentation head. Table 2, Figures 10 and 11 summarize the overall performance of each architecture. The results show that the number of trainable parameters does not necessarily correlate with training time because 2D convolution at higher resolution is more computationally intensive. This is also the case in the reconstruction phase when the reconstruction time in U-Net with depth = 3 is 2.34 s per slice, which is greater than that of depth = 4 and depth = 5 by 43% and 34%, respectively. The noticeable difference between the train/validation mIoU and test mIoU for depth = 4 can be a sign of overfitting on both the train and validation sets. Using cross-validation techniques such as 10-fold cross-validation can minimize this problem.

**Table 2.** U-Net segmentation performance at different levels.

| Depth | Decoder Channels | Number of Parameters | Training Time [min] | Reconstruction Time [sec/slice] | Overall Performance on Test Set [mIoU ± σ] | Best Performance on Test Set | | |
|---|---|---|---|---|---|---|---|---|
| | | | | | | Train mIoU | Val. mIoU | Test mIoU |
| 5 | (256, 128, 64, 32, 16) | 14,328,354 | ~25 | 1.64 | 0.7933 ± 0.0196 | 0.7992 | 0.9050 | 0.8156 |
| 4 | (256, 128, 64, 32) | 13,344,770 | ~25 | 1.75 | 0.7944 ± 0.0071 | 0.9078 | 0.9240 | 0.8090 |
| 3 | (256, 128, 64) | 12,838,338 | ~50 | 2.34 | **0.8016 ± 0.0127** | 0.8593 | 0.8409 | **0.8181** |

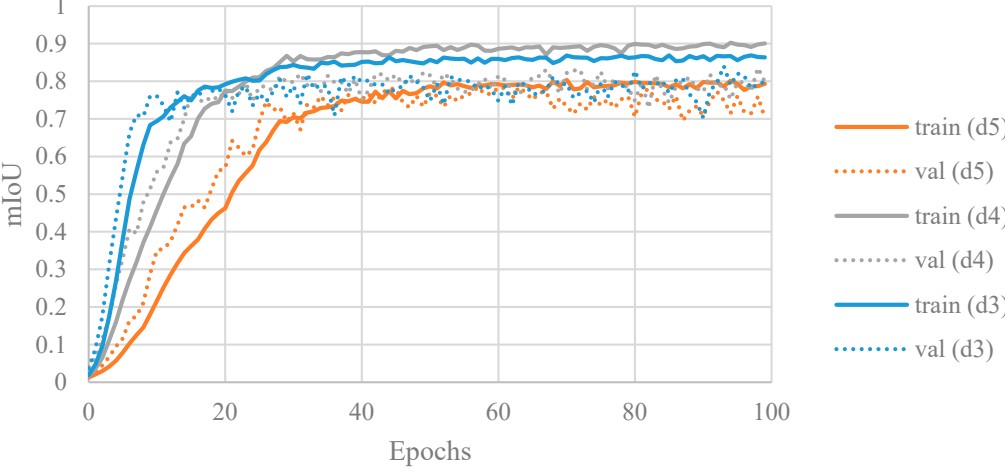

**Figure 10.** Training and validation evolution for each depth. The shallow network can converge faster than deeper networks.

Since U-Net with depth = 3 showed better performance and has a faster convergence rate (Figure 10), the changes in validation and training mIoU for the best three-layer model are illustrated in Figure 12. The effect of multiple random weight initialization on training evolution of this architecture on the training set is shown in Figure 13 numerically and visually. The stability of the minimum, maximum, and average values of mIoU in training curves (Figure 13) indicate that converging to local minima is unlikely and the network is able to converge to a narrow interval of mIoU values from randomly sampled initial weights.

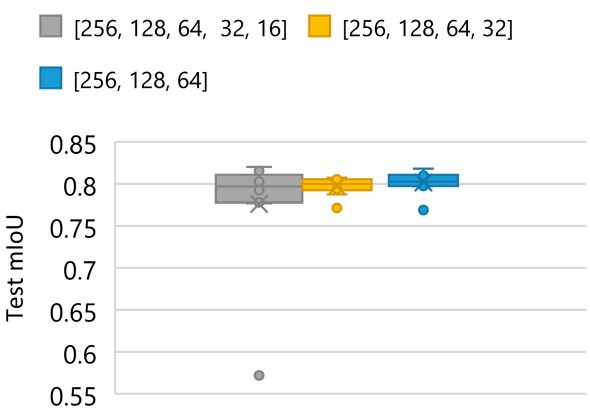

**Figure 11.** Box and whisker plot of mIoU on test set for different depth of U-Net.

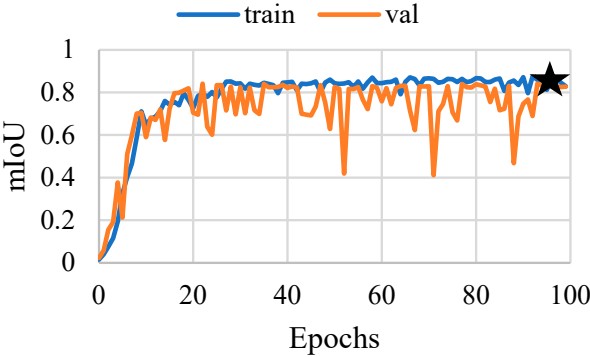

**Figure 12.** Training and validation mIoU at each epoch for U-Net with depth = 3; The best validation mIoU is achieved at epoch 97 with value of 84.09% (shown with the star symbol).

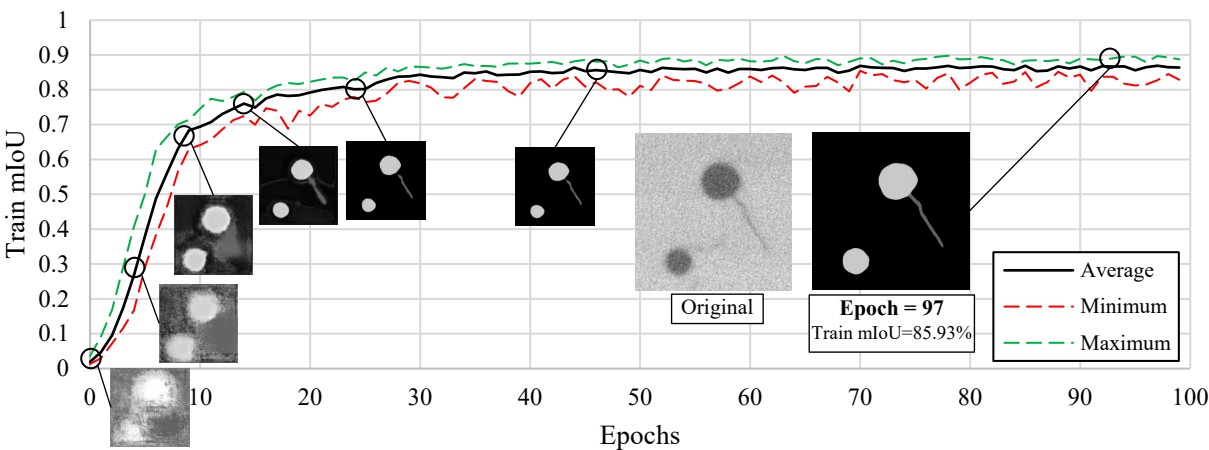

**Figure 13.** Training mIoU of U-Net (depth = 3) at each epoch; training is repeated 10 times and minimum, average, and maximum values of mIoU on training set are calculated.

Possible explanations for obtaining better results for shallower networks could be higher chances of overfitting at deeper (i.e., more complicated) networks. Moreover, according to Figure 5, severe downscaling may destroy the features inside the patches and cause the information to be lost completely. Therefore, more layers and lower resolutions not only do not affect performance but also may mislead the optimizer in finding the correct features.

After training the architectures, all the slices of the tomography dataset are segmented using the one that had the best test mIoU (i.e., U-Net depth = 3). The original image and the segmented image are shown in Figure 14. It should be noted that this image has never been seen by the network during any of training, validation, or previous testing phases.

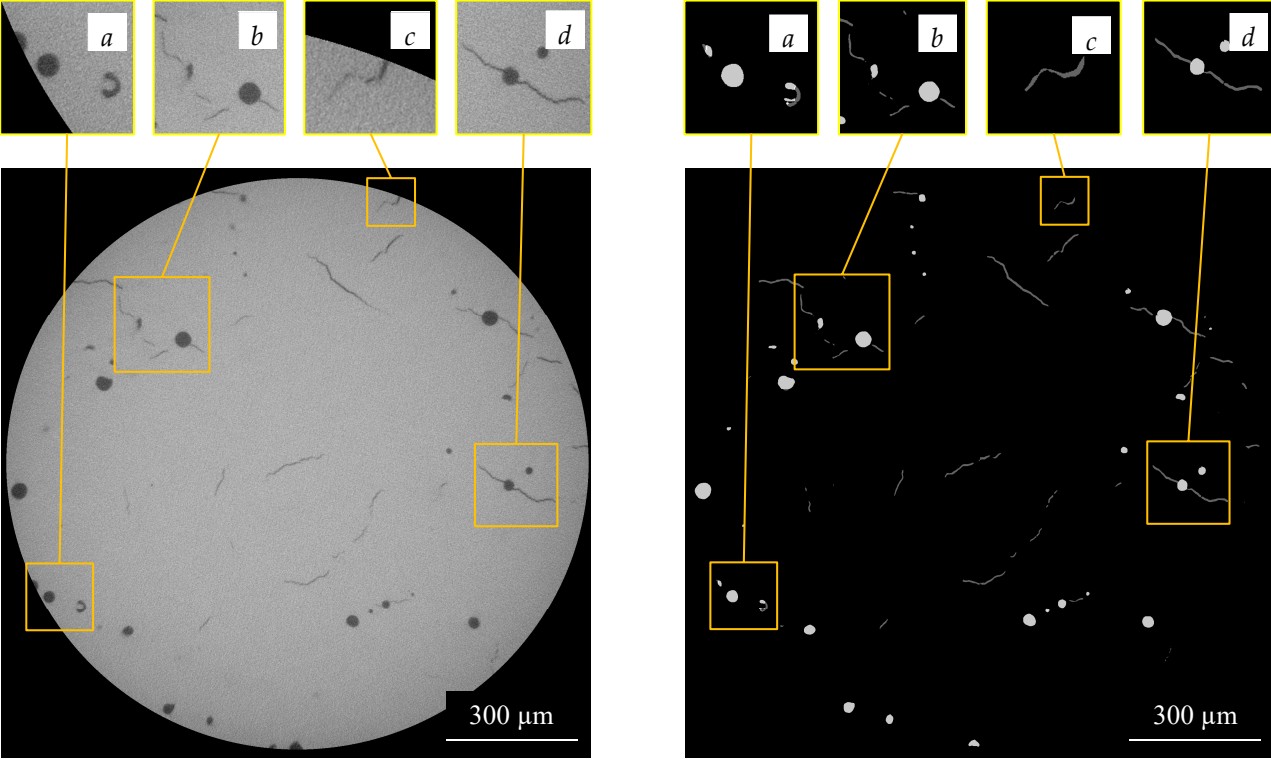

**Figure 14.** Segmentation performance of the network for a new slice of tomography dataset. (**left**): original image, (**right**): segmented map by AI generated in about 2.3 s. (**a**) detecting pores intersected with background and non-circular pores, (**b**) detecting a cluster of pores and cracks (**c**) detecting a crack intersected with background (**d**) detecting connected pores and cracks.

It can be seen in Figure 14b that the AI agent has successfully segmented the cluster of cracks and pores based on their morphology and contrast in the original image. Figure 14c shows that the trained network is also doing a great job in identifying the crack on the edge of the scanning area. This capability can be extremely advantageous in detecting cracks on the surface of the part, as many of the fatigue-induced failures are caused by crack initiations from the surface of the specimen. The network's ability to differentiate between cracks and pores is showcased in Figure 14d, as it has successfully segmented the cracks and pores even though they are connected and have almost the same contrast. This level of discrimination power is nearly impossible using typical thresholding methods. However, based on what is shown in Figure 14a, the network is having trouble with the correct segmentation of near-the-edge pores and the narrow ones that may look like thick cracks.

It takes about 72 min for the single NVIDIA GeForce GTX 1650 GPU to segment 1879 images. A 3D reconstruction representation of the segmented defects along with the original dataset is shown in Figure 15. A rendered video of the crack and pore distribution is also available in the Supplementary Materials.

The obtained morphology and spatial distribution of pores and cracks plays a crucial role in determining the static strength and fatigue performance and the direction of possible failure. In a data mining approach, this information can be useful in finding the correlation between the process parameters (laser power, direction, speed, etc.) at each time and adjusting them accordingly to minimize the presence of such defects, which is left for future research.

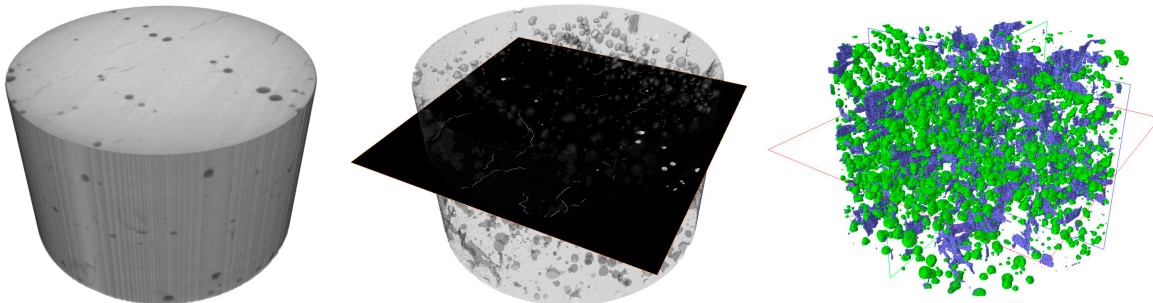

**Figure 15.** Three-dimensional reconstruction of the tomography dataset in Avizo; (**left**): original dataset, (**middle**): segmented dataset using AI with cross section, (**right**): quantified distribution of cracks and pores (blue: cracks, green: pores).

Table 3 compares the duration of each stage for manual and automated segmentation, which points out that the performance of the AI architecture is enormously better than human experts in terms of speed. After breaking down the typical stages of automated segmentation and their duration, segmenting 1000 images using ML will take orders of 10 h, while it takes weeks for a person to fulfill the same task.

**Table 3.** Comparison between manual and automated segmentation in terms of duration for 1000 slices.

| Segmentation Method | Manual Segmentation | Preprocessing | Training | Reconstruction | Total |
|---|---|---|---|---|---|
| Manual | 1000 h (>5 weeks) | N/A | N/A | N/A | 5 weeks |
| Automated (Supervised) | 1–2 h (for 2 training slices) | 2-4 h | 5-15 h | 1 h | Less than 20 h |

The first and most accessible metrics that can be calculated from the 3D reconstructed volume are the crack and pore volume fractions, which are summarized in Table 4. With each pixel being assigned to a class in the sample (crack, pore, background, etc.), all the defect-related measures, including—but not limited to—pore/crack density, volume fraction, orientation, morphology, etc., are also readily available.

**Table 4.** Defect characterization inside the part.

| Defect Type | Volume Fraction |
|---|---|
| Crack | 0.00426696 |
| Pore | 0.0222326 |

Based on the Table 4 results, the porosity of the selected volume is calculated to be 2.65%, which is about 29% lower than the experimental value for porosity calculated by Archimedes principle in Section 2. This discrepancy roots from the aggregation of error in the following stages:

1. Error in experimental porosity measurement using Archimedes principle;
2. Voids not being captured in XCT;
3. Error in manual segmentation for generating training data;
4. Network being incapable of performing the segmentation correctly.

Based on the comprehensive investigation conducted at NIST [51], XCT generally predicts lower values of porosity compared to the Archimedes principle, especially for samples with higher porosities. Water infiltration caused by surface pores and cracks (which is the case here) can contribute to erroneous measurement in Archimedes method. On the other hand, XCT is not capable of capturing the voids smaller than the voxel size,

which leads to *underestimating* the porosity. Knowing that, now we only focus on identifying the error sources in manual image segmentation and the learning capability of the network for segmenting the pores and cracks correctly.

In terms of segmentation accuracy, an interesting behavior is noticed while investigating the segmented maps during training. As illustrated in Figure 16, at some point AI starts to outperform the human expert (which is represented by this ground-truth image), but obtains lower mIoU scores and will be enforced by the optimizer to reconstruct the exact same map as the ground truth. This is mainly because of human error in the manual image segmentation stage and the limited tools, energy, and patience that human beings have. Possible solutions to this issue are uncertainty analysis of the manually segmented labels, semi-supervised, and zero-shot (unsupervised) learning approaches.

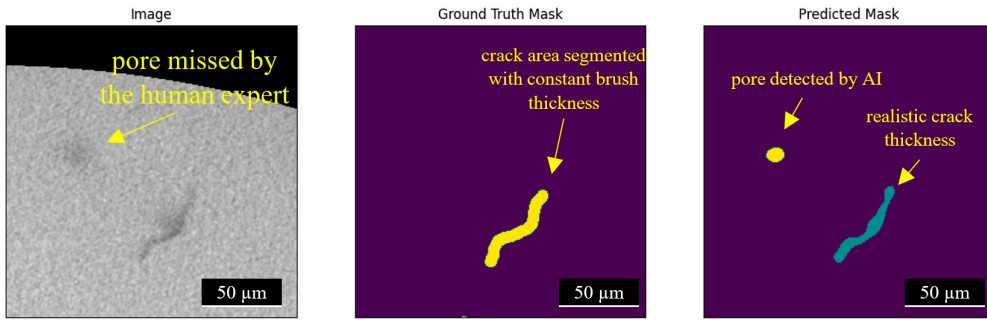

**Figure 16.** AI outperforming human expert (ground truth mask) during training.

The results make it clear that pixel-level segmentation is extremely prone to human error. Full reliance on erroneous manually segmented areas and forcing the network to reconstruct and resemble them prevent leveraging the full discrimination capacity of the AI architectures. There are many examples of images segmented by AI with lower mIoU scores compared to the target value, which should override the manually segmented image when being investigated for the second time, even based on the opinion of the expert who performed the manual segmentation in the first place. This brings up the necessity of uncertainty analysis of manually segmented images and applying zero-shot learning, and researchers are encouraged to address them in the future.

## 6. Conclusions

Automated evaluation and quantification of a large tomography dataset of a Ni-939 sample fabricated using the LPBF additive manufacturing method is studied. The porosity of the sample is estimated using the Archimedes principle and compared to XCT, which determines the upper and lower band for the ground-truth porosity value. The results show that AI architectures are able to mimic human experts up to 82%. For detecting two classes of typical defects (i.e., cracks and pores) inside the AM sample, deeper networks do not provide a better performance based on the results. Possible reasons are the increased chance of overfitting in more complicated networks and the simplicity of the features inside the dataset. Multiple random weight initialization proved to be an effective training practice in avoiding local minima. Although depth optimization and random weight initialization enhanced the total segmentation performance, for further improvement of the segmentation accuracy, future research should be focused on addressing the uncertainty of manually segmented images and minimizing human error in the few-shot approach or eliminating it entirely by adopting a zero-shot learning strategy. This paper has provided the required methodology and information for making a correlation between processing parameters and defect distribution of the fabricated part, which eventually can be used for fatigue life estimation simulations.

**Supplementary Materials:** The following supporting information can be downloaded at: https://www.mdpi.com/article/10.3390/jmmp6060141/s1, Video S1: Three-dimensional rendering of defect distribution.

**Author Contributions:** S.N. was in charge of writing the manuscript, data collection, model development, and evaluation. H.G. was in charge of sample preparation and processing. X.L. supervised the image processing and computer vision architecture and modeling. L.G.B. supervised the X-ray tomography process. H.W. performed LPBF sample preparation and material characterization. S.G. supervised the entire data-driven material development approach. All authors wrote the abstract and conclusion sections and reviewed the manuscript. All authors have read and agreed to the published version of the manuscript.

**Funding:** This work is supported by the U.S. National Science Foundation under grant number OIA-1946231 and the Louisiana Board of Regents for the Louisiana Materials Design Alliance (LAMDA).

**Data Availability Statement:** The developed repository can be accessed using the following GitHub link, accessed on 21 October 2022: https://github.com/LSU-LAMDA/crack-pore-detection.

**Conflicts of Interest:** The authors declare no conflict of interest. The funders had no role in the design of the study; in the collection, analyses, or interpretation of data; in the writing of the manuscript; or in the decision to publish the results.

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
