# Peer review of "Automated Defect Analysis of Additively Fabricated Metallic Parts Using Deep Convolutional Neural Networks"

_jmmp, doi:10.3390/jmmp6060141_

Round 1
Reviewer 1 Report
The authors used the U-Net architecture to automatically evaluate and quantify the tomography data of a Ni-939 sample fabricated by LPBF additive manufacturing. The trained network is able to segment the unbalanced classes of pores and cracks with a mean Intersection over Union (mIoU) value of 82%. The paper is well constructed. The experimental results are nice and well interpreted. This work is recommended for publication after a minor revision due to the following questions:
1. Page 7, line 248. It should be Figure 7 instead of Figure 5.
2. Page 8, line 266. Can authors discuss whether two images are enough for the model training?
3. Figure 17, the right one. What the blue and green colors represent (i.e., pore or crack) should be indicated in the figure or figure caption.
4. Page 14 line 437. Please double check whether the water infiltration overestimates or underestimates the porosity value in Archimedes method. In my opinion, it underestimates.
Author Response
Thank you for your valuable feedback and comments. Please read the following responses related to your remarks:
1- Revised.
2- Added to the following explanation: "which include all the features and shown to be sufficient for training in the results section"
3- Included in the figure caption.
4- Revised. Archimedes also underestimates, but NIST paper showed that generally, XCT shows lower values.
Reviewer 2 Report
Thanks for inviting me to evaluate the article titled “Automated Defect Analysis of Additively Fabricated Metallic Parts using Deep Convolutional Neural Networks”. This work focus on automated defect analysis of metallic parts fabricated by Laser Powder Bed Fusion (LPBF). The internal defects will greatly affect the performance of AM parts, so having a thorough understanding of these defects can help researchers to find appropriate processing parameters to decrease the internal defects. In this work, to detect defects more efficiently, a segmentation architecture based on U-Net is applied to identify the cracks and pores in CT images of LPBF samples, and three bottlenecks (network depth, random weight initialization, errors in manually labeled data) of U-Net are also investigated. The result shows that the trained network is able to segment defects with MIoU value of 82% and has reduced the characterization time remarkably compared to manual methods. This work also analysis the discrepancy roots from the aggregation of error in detail, which could provide guidance for the application of U-Net in automated defects detection. This is a well-done work and some novel ideas are interesting for readers. A few revisions and some suggestions are list below.
1. In introduction, Figure1 (The workflow of a typical data-driven model) and Figure 2 (Typical workflow of automated image segmentation frameworks) seem to be redundant, a concise description is enough for the background of data-driven method, or you can take it into supplementary materials. And the introduction of relevant works (references) in automatic defect detection is not enough, you may add some details about the progress of this field in recent years.
2. Effect of network depth and random weight initialization are two main parameters investigated in this work. However, the degradation of U-Net when the depth increases is a widely accepted phenomenon and the random weight initialization is a commonly used network initialization method. And the effect of random weight initialization is not evident in this article, may add some comparations to other initialization method. I think the authors could pay more attention to data characteristics in this field instead of hyperparameters of networks, e.g., the resolution of patches, the contrast of CT images and the thickness of cracks, etc.
3. There is a mispatch in the introduction and the discussion/conclusion. The authors mentioned that they are in better position to assess the strength and weakness of U-Net, but I do not see any discussion on the strength/weakness of U-Net in following paragraphs. “This paper tries to reveal the ultimate of U-Net”, but only three bottlenecks are studied, is the ultimate of U-Net truly reached? The authors could do some analysis on the strength/weakness of U-Net based on this work, and some strategies that would improve the accuracy of U-Net in automatic detect defects could be properly summarized in conclusion.
Author Response
Thank you for your valuable feedback and comments. Please read the following responses related to your remarks:
- The first two figures are removed. The last paragraphs of the introduction section are rewritten.
- I added more discussion regarding this item:
- About the effect of random initialization: In section 5, results and discussion: "The stability of the minimum, maximum, and average values of mIoU in training curves (Figure 13) indicate that converging to local minima is unlikely and the network is able to converge to a narrow interval of mIoU values from randomly sampled initial weights."
- About data characteristics:
- In section 4.2 data preparation: The selected images contain all the color contrast and morphological features that need to be learned by the network.
- In section 4.2 data preparation: Preliminary test runs show that this patch size can encompass the morphological characteristics of both crack and pore features. A larger patch size will not produce enough training data while very small ones will not be able to represent the features properly.
- The mentioned sentences are revised:
- In the Introduction section: "U-Net is the basis for many new architectures for image segmentation [27-35]. The goal of this paper is to comprehensively identify any unique situations that either U-Net cannot handle, or needs to be addressed in a different way other than using more complicated architectures."
- In the Introduction section: "As we develop an understanding of the shortcomings of the current automated defect segmentation process with U-Net, we are in a better position to pinpoint the critical stages and assess the strengths and weaknesses of the new image segmentation architectures, if ever needed to use a more complicated one."
Reviewer 3 Report
The presented investigation tackles a relevant problem in additive manufacturing, however, from my perspective there are some critical issues that must be addressed.
Major:
- Please point out the novelity of the presented approach, since there are a myriade of investigantions and publications following the exact same approach using ANNs for defect analysis in additive manufacturing
- Here, I also feel that the state of the art section should be rewritten and consider more of this previous work.
- The variation and actual setting of the hyperparamter variation and the adaption of the architectrue should be discussed in more detail, especially with regard to the taken values.
- Individual human experts should not be used as a measure of quality for Machine Learning models
Minor:
- Font sizes in figures should be adapted, since most of the text is not readable.
- Quality of the figures should be reconsidered.
- Numbering and referencing of the figures is not correct.
Author Response
Thank you for your valuable feedback and comments. Please read the following responses related to your remarks:
Major:
- The purpose of the research is to thoroughly investigate a well-known architecture and systematically find the bottlenecks for improving the accuracy. We showed that the major bottleneck is the "uncertainty of the labeled data" and without addressing this problem first, designing new architectures will not lead to substantial improvement. This has been stated in the abstract, introduction, results, and convolution sections.
- The few last paragraphs of the introduction section are rewritten, per the reviewer's suggestion.
- I'm not sure if I understood the reviewer's comment correctly. However, the discussion about selecting the hyperparameters is added to the manuscript.
- In section 4.2 data preparation: The selected images contain all the color contrast and morphological features that need to be learned by the network.
- In section 4.2 data preparation: Preliminary test runs show that this patch size can encompass the morphological characteristics of both crack and pore features. A larger patch size will not produce enough training data while very small ones will not be able to represent the features properly.
- We asked 5 other researchers to do the same task and the results were worse, if not the same. The reason being is the limited tool, energy, and patience for such an amount of data, which is stated in the paragraph before figure 16.
Minor:
- Figure 1 and Figure 2 are removed, because of another reviewer's comment. The rest are modified to be readable.
- The figures related to results are mostly vector. The first two figures are removed.
- Corrected.
Round 2
Reviewer 3 Report
The raised issues have been included in the revised paper that now can be accepted for publication in my view.